# Home Parenteral Nutrition in Patients with Intestinal Failure: Possible Undetected Complications

**DOI:** 10.3390/nu11030581

**Published:** 2019-03-09

**Authors:** Lidia Santarpia, Giulio Viceconte, Maria Foggia, Lucia Alfonsi, Grazia Tosone, Luigi Camera, Maria Carmen Pagano, Giuseppe De Simone, Franco Contaldo, Fabrizio Pasanisi

**Affiliations:** 1Internal Medicine and Clinical Nutrition, Department of Clinical Medicine and Surgery, Federico II University of Naples, 80131 Naples, Italy; lucia.alfonsi@unina.it (L.A.); carmenpag@libero.it (M.C.P.); contaldo@unina.it (F.C.); pasanisi@unina.it (F.P.); 2Infectious Disease, Department of Clinical Medicine and Surgery, Federico II University of Naples, 80131 Naples, Italy; giulio.viceconte@gmail.com (G.V.); mariafoggia@alice.it (M.F.); tosone@unina.it (G.T.); 3Radiology, Department of Advanced Biomedical Sciences, Federico II University of Naples, 80131 Naples, Italy; camera@unina.it; 4Anesthesiology and Intensive Care Unit, Department of Neurosciences, Federico II University, 80131 Naples, Italy; gdesimone@inwind.it

**Keywords:** central venous catheter related bloodstream infection, home parenteral nutrition, septic pulmonary embolism, parenteral nutrition

## Abstract

Background: Septic pulmonary embolism (SPE) may be a frequently undetected complication of central venous catheter (CVC)-related bloodstream infections (CRBSIs). Materials and Methods: The incidence of SPE was evaluated in a cohort of non-oncological patients on home parenteral nutrition (HPN) who were hospitalized for a CRBSI from January 2013 to December 2017. The main clinical, microbiological, and radiological features and the therapeutic approach were also described. Results: Twenty-three infections over 51,563 days of HPN therapy were observed, corresponding to an infection rate of 0.45/1000. In 10 out of the 23 cases (43.5%), pulmonary lesions compatible with SPE were identified. Conclusion: Our results demonstrated that a CRBSI can produce asymptomatic SPE with lung infiltrates in 43.5% of the cases, suggesting the need to check for secondary lung infections to choose the most appropriate antimicrobial therapy.

## 1. Introduction

Home parenteral nutrition (HPN) is a life-saving treatment for patients with chronic benign intestinal failure [1,2]. Central venous catheter (CVC)-related bloodstream infections (CRBSIs) represent one of the most serious and common complications of long-term HPN [2,3,4,5]; in some cases, septic pulmonary embolism (SPE) has been detected. Indeed, infected thrombi could move from the primary infectious site to the lung through the pulmonary artery [5].

Furthermore, on these occasions, patients may present non-specific clinical symptoms, and consequently, the diagnosis of this disorder may often be delayed, resulting in a poor treatment outcome [6,7].

In fact, if not promptly diagnosed, SPE can prolong the duration of treatment and hospitalization, particularly when the selected antibiotic therapy for CRBSI treatment may have an uncertain efficacy in treating lung infections, as in the case of Daptomycin (DAP) [8,9,10,11,12,13].

This study aims to retrospectively evaluate the incidence of CRBSI and SPE in a cohort of non-oncological patients on HPN followed at the Internal Medicine and Clinical Nutrition Unit, Department of Internal Medicine, Federico II University Hospital, Italy.

The main clinical, microbiological, and radiological features of SPE secondary to CVC infection and the prescribed antibiotic treatment are also described.

## 2. Patients and Methods

All patients on HPN carrying a CVC or a peripherally inserted central venous catheter (PICC) from January 2013 to December 2017 were hospitalized for a CRBSI and evaluated.

Only adult patients with non-oncological gastrointestinal diseases who were on HPN through a catheter for more than two weeks were included in the study.

The exclusion criteria were as follows: HPN for less than two weeks, oncological diseases, less than 18 years old, and a lack of central venous access.

For each patient, information was collected on age, gender, underlying disease, relevant comorbidities, implanted CVC type and time from implantation, number of CRBSIs, type of infecting agents, days of catheterization, number of infusions per week, previous CVC infections, clinical condition at admission (e.g., blood pressure, pulse rate), and hemato-biochemical exams (white blood cell count, sedimentation rate (RS), C-reactive protein (CRP), procalcitonin). Type and modality of antibiotic treatment performed according to the antibiogram and clinical outcome were also recorded. CRBSI were diagnosed according to the IDSA Guidelines for Diagnosis and Management of Intravascular Catheter-related Bloodstream Infection and ESCMID Guidelines for the Diagnosis and Treatment of Biofilm Infections [14,15]. In detail, diagnosis of CRBSI requires that the same organisms grow from at least one percutaneous blood sample culture and from the catheter tip or that two blood samples for culture be obtained (one from a catheter hub and one from a peripheral vein) that meet the CRBSI criteria for quantitative blood cultures. This criteria foresees the growth of microbes in blood drawn from a catheter hub at least 2 h before growth is detected in blood samples obtained from a peripheral vein with no other apparent source of infection. At the time of CVC infection diagnosis, transthoracic echocardiogram, venous vascular echo-color-Doppler and chest high-resolution CT scans were performed in all patients. An endoscopic esophageal echocardiogram was made only in cases of dubious transthoracic echocardiograms.

Before starting HPN, all patients and/or their caregivers received oral and written instructions from the nutrition team on the CVC aseptic management and on to how to recognize infectious (and non-infectious) complications. According to the ESPEN guideline recommendations [16] central venous access devices were routinely flushed and locked with saline solution when not in use.

During HPN infusion, a hydrophobic antibacterial filter with a porosity of 0.4 microns was routinely used. The antibacterial filter was placed on the air intake valve.

The chest CT scan images were retrospectively evaluated at the time of the study by an expert radiologist for eventual SPE compatibility.

CT findings were considered SPE compatible if the following pre-specified criteria by Cook et al. [17] were present: (a) focal or multifocal lung infiltrates compatible with septic lung embolism; (b) presence of an active extra-pulmonary infection as a potential embolic source; (c) exclusion of other potential explanations for lung infiltrates; and (d) resolution of lung infiltrates with appropriate antimicrobial therapies.

The images were assessed for the following radiological patterns: non-nodular infiltrates, nodules with or without cavities, unilateral or bilateral abnormalities, sub-pleural opacities, pleural involvement, and pleural effusions.

The study was approved by the Local Ethics Committee of the Federico II University Hospital (prot. n. 37/11) and informed consent was obtained from all participants.

## 3. Statistical Analysis

Continuous variables were compared by the Kolmogorov–Smirnov test for non-normally distributed variables. Categorical variables were evaluated by using logistic regression. The results are expressed as median + IQR (continuous variables) or as percentages of the group from which they were derived (categorical variables). Differences among variables were considered statistically significant for *p* values < 0.05. All statistical analyses were performed by using the IBM SPSS Statistics^®^ program, version 20 (IBM Corp., Armonk, NY, USA).

## 4. Results

From January 2013 to December 2017, 181 patients (81 males, 100 females, mean age 54 ± 18 (median 55; range 13–90 years)) suffering from a non-oncological gastrointestinal disease on HPN were recruited for a total of 51,563 days of therapy (mean 285 ± 405; median 120; range 24–1598).

In all, 23 infections were recorded with a rate of 0.45/1000 days of HPN therapy.

The 23 infections occurred in 15 patients: specifically, one patient had four infections over four years, two patients had three infections over three years, one patient had two infections in one year, and the remaining 11 patients had one infection. If in the same patient, a separate event occurred at least three months after the previous one, it was considered as a new CRBSI episode.

The characteristics of the 15 hospitalized patients diagnosed with CRBSI (7 M, 8 F, median age 57 years, IQR 23–79) are summarized in Table 1.

Thirteen of the 15 (86.7%) patients had already had a previous CVC insertion and at least one previous CRBSI diagnosis. In seven of 13 (53.8%) patients, the previous event occurred on the same CVC present at the time of hospitalization.

As in the past and in the present, a consolidated protocol was used. Following the culture results, infected catheters were treated with systemic intravenous antibiotic and “lock therapy”. The latter consisted of the injection of the antibiotic with the lowest minimal inhibiting concentration (MIC), diluted with 5–7 mL of saline solution (which completely fills the catheter) directly into the catheter once or twice daily (according to its half-life) for two weeks. For systemic intravenous therapy, generally a second antibiotic is chosen. The infusion of the nutritional mixture through the catheter is restored at the fourth day of antibiotic therapy. Reoccurrence of fever during antibiotic “lock” administration, at catheter use restoration, or at the end of the antibiotic treatment, is considered therapy failure and an indication for CVC removal [3,4,8].

Significant comorbidities are reported in Table 1. No patient had a respiratory comorbidity.

At admission, patients presented only symptoms related to CRBSI: increased body temperature with shivering during parenteral nutrition infusion (20 cases) or prolonged mild fever shiver-free, not strictly related to the nutritional infusion (three cases). No patients had specific symptoms related to lung involvement such as cough, chest pain, or dyspnoea.

At diagnosis, all CRBSI patients underwent a high-resolution chest CT scan.

In 10 of the 23 cases (43.5%; 5 patients), pulmonary lesions compatible with septic pulmonary emboli were first identified by a radiologist and confirmed by a second expert at the time of the study. Isolated pathogens, their resistance profiles, rates of lung involvement, and antibiotic combinations used are shown in Table 2.

All CRBSIs were treated with targeted antibiotic therapy, generally for two weeks, after the results of the antibiogram.

### Outcomes

In 16 of the 23 (69.6%; 11 patients) cases of CRBSI, CVC was removed according to the guidelines, particularly in 14 cases due to an infection of a short-term catheter (12 PICCs and 2 non-tunneled CVCs) and in one case due to a *C. parapsilosis* infection.

In 7/8 infections (five patients) ALT was successful in restoring the infected long-term catheter.

All performed transthoracic echocardiograms were negative for valvular vegetations.

In 6/23 (26.1%; six patients) cases, the venous echo-color-Doppler showed peri-catheter thrombosis. In four cases, the thrombus was on the vessel wall, surrounding the catheter external surface and partially occluding the vein; in two cases, the thrombus completely occluded the catheter tip.

In these conditions, all catheters were PICCs, and the infecting agents were as follows: one from *S. epidermidis*, two from *S. aureus*, one from *S. haemolyticus*, one from *Candida parapsilosis*, and one from *Enterobacter cloacae* + *Acinetobacter*.

Median hospitalization was 15.5 days (IQR 10.25–28).

Gram-positive bacteria were more frequently associated with lung involvement (Table 2), but the difference was not statistically significant. Table 3 shows the CT findings for the 10 CRBSIs with lung involvement. The CT images in cases of SPE varied from one patient to another, presenting nodules, patchy infiltrates, cavity and pleural effusion, individually or in association.

No correlation was found between the patient characteristics listed in Table 1 and lung infection risk.

After 15 days of antibiotic therapy, a post-treatment chest CT scan was performed in all cases with lung involvement; in 7/10 cases (five patients), we observed complete resolution of septic lesions, and in 3/10 cases (three patients), the resolution was only partial. In these last three cases, a third chest CT scan was performed after an additional month, showing complete resolution of the pulmonary frame (see an example in Figure 1).

No significant differences in the clinical course were observed in SPE patients when compared to those with CRBSI alone. No statistically significant correlations were found between SPE and clinical or microbiological variables.

## 5. Discussion

CRBSI is a severe and frequent complication of HPN and a common cause of hospitalization [18].

SPE is a possible complication of CRBSI and is caused by the migration of infectious agents to the lung through the pulmonary artery [19,20].

Our study demonstrated that CRBSI can produce asymptomatic SPE with lung infiltrates, which resulted in 43.5% of cases (five patients). SPE completely lacks clinical symptoms for pulmonary involvement (dyspnoea, cough, shortness of breath, and chest pain); therefore, it could be disregarded, negatively influencing the patient’s outcome. If not promptly diagnosed, SPE can prolong treatment and hospitalization, particularly when the selected antibiotic is not specifically indicated for lung infections, as in the case of Daptomycin (DAP).

DAP is a bactericidal lipopeptide active against CoNs and MRSA and has a high penetration rate into various tissues [5,7,8,9,10,11]. Unfortunately, it is still unknown if its efficacy in treating septic thromboemboli in lung vessels is as successful as in other extrapulmonary sites [12,13].

In MR Gram-positive bacteria SPE, other antibiotics could be preferred according to the antibiogram, patient clinical conditions, and possible drug side effects.

In our experience, the presence of SPE was suspected in those cases with persisting fever despite antibiotic therapy and confirmed by a chest CT scan, thus increasing antibiotic therapy enhancement or modification with a consequent extension of therapy.

In other cases, lung involvement during a CRBSI was accidentally detected during an angio-CT scan performed on the suspicion of deep vein thrombosis (superior cava or subclavian veins) and/or in the study of the venous system for an eventual new CVC implantation.

In SPE found by CT scan, individual patients have more than one imaging manifestation simultaneously (nodules, patchy infiltrates, and cavities), in particular, we wanted to underline excavations in one episode of SPE by *Kocuria* aside from the two cases by MRSA.

Limitations of this study include the relatively small patient sample and the selection bias, being a single-center study. Despite these shortcomings, this study suggests the need to consider lung infections associated with CRBSs due to their relatively high frequency, and to the choice of the appropriate antimicrobial therapy. In case of SPE, chest radiography had low sensitivity and specificity [19,20], and consequently a chest CT scan could be useful, at least in selected cases.

In this study, thanks to the early detection of lung involvement and the use of the most suitable antibiotic therapy, patients with and without SPE had the same duration of treatment and hospitalization.

In conclusion, in the present study, the chest CT scan showed septic involvement of the lungs in 43.5% of CRBSI patients, in the absence of clinical signs or symptoms.

We speculated that in CRBSI patients with a high risk of developing severe infections, a chest CT scan may contribute to the detection of possible septic pulmonary involvement, thus preventing treatment failure and other potential complications. These preliminary observations need to be confirmed in larger multicenter studies.

## Figures and Tables

**Figure 1 nutrients-11-00581-f001:**
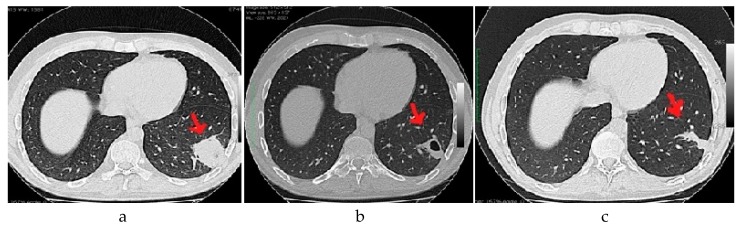
Chest CT scan demonstrating the evolution of an infiltrate in the inferior lobe of left lung due to septic pulmonary embolism by S. aureus infection (see Table 3, patient C). (**a**) At the diagnosis; (**b**) After 15 days antibiotic therapy; (**c**) Follow up one month after the end of antibiotic therapy.

**Table 1 nutrients-11-00581-t001:** Baseline characteristics of the 15 patients with one or more episodes of CRBSI (total infections = 23).

**Patients’ Characteristics**	
Sex, *n* (%)	
Males	7/15 (46.7%)
Females	8/15 (53.3%)
Age (years) median (IQR)	57 (23–79)
Days of catheterization, total, median (IQR)	14,552, 1000, (90–2060)
Days of HPN, median (IQR)	13,211, 870 (90–1526)
**HPN Frequency, Days (%)**	
7 days per week	12/15 (80%)
5 days per week	2/15 (13.3%)
3 days per week	1/15 (6.7%)
**Previous CVC Insertion, *n* (%)**	13/15 (86.7%)
**Previous CRBSI, *n* (%)**	13/15 (86.7%)
**Previous CRBSI on the same CVC, *n* (%)**	7/13 (53.8%)
**Intestinal Diseases, *n* (%)**	
Intestinal infarction	6/15 (35.3%)
Crohn disease	4/15 (26.7%)
Intestinal volvulus	2/15 (11.8%)
Radiation enteritis	2/15 (11.8%)
Whipple resection	1/15 (6.7%)
**Length of Residual Bowel, *n* (%)**	
<50 cm	8/15 (53.3%)
50–100 cm	5/15 (33.3%)
>100 ≤ 200 cm	2/15 (11.8%)
**Ileostomy, *n* (%)**	5/15 (33.3%)
**Length of Colon in Continuity, *n* (%)**	
70% (no ileocecal valve)	7/10 (70%)
100%	3/10 (30%)
**Comorbidities, *n* (%)**	
Respiratory diseases	0
Chronic deep vein thrombosis	6/15 (35.3%)
Chronic kidney disease	2/15 (11.8%)
Hypertension	4/15 (26.7%)
Peripheral artery disease	1/15 (5.9%)
**Type of Infected Catheter, *n* (%)**	
Hickman tunneled (long term)	10/23 (43.5%)
PICC (short term)	11/23 (47.8%)
Non-tunneled CVC (short term)	2/23 (8.7%)
**Reported Symptoms**	
Strictly related to CRBSI	20 (87%)
Not related to CRBSI	3 (13%)
**Positive Chest CT Scan, *n* (%)**	10/23 (43.5%)
**Antibiotic Lock Therapy, *n***	8 (7 successful)
**CVC Removal, *n* (%)**	15/23 (65.2%)
**Hospital Length-of-Stay (Days) Median (IQR)**	15.5 (10.25–28)

HPN = Home Parenteral Nutrition; CVC = Central Venous Catheter; CRBSI = Catheter-related bloodstream infection.

**Table 2 nutrients-11-00581-t002:** Infecting agent, antibiotic therapy used, number of treated patients, and lung involvement of the 23 episodes of CRBSI.

Infecting Agents Type and (*n*)	Drug Combinations	Treated Patients (*n*)	CT-Scan Positive for SPE (*n*)
MRSE, vancomycin MIC > 1 (1)	Tigecyclin + Teicoplanin	1	0
MRSE (4), CoNs (4)	Daptomycin + Rifampicin	8	0
MRSE vancomycin MIC > 1 (2)	Linezolid + ciprofloxacin	2	2
Methicillin-resistant S. aureus (MRSA) (2)	Linezolid + Piperacillin-tazobactam	2	2
Methicillin-resistant S. aureus (MRSA) (1)	Levofloxacin + Ceftobiprole	1	1
Kocuria kristinae (1)	Linezolid + Ciprofloxacin	1	1
S. haemoliticus + Citrobacter freundii (1)	Levofloxacin + Ceftriaxone	1	1
S. epidermidis (1)	Teicoplanin + Tigeciclin	1	1
S. chromogenes (1)	Piperacillin-tazobactam + Rifampicin	1	1
EBSL-producing Pseudomonas aeruginosa (1)	Gentamicin + Ceftazidime	1	1
Proteus mirabilis (1), Enterobacter cloacae (1), CoNs (1), MRSE + Enterococcus fecalis (1)	Others	4	0

SPE = Septic Pulmonary Embolism; ESBL = Extended spectrum beta-lactamase; MRSE = Methicillin-resistant Staphylococcus epidermidis; CoNs = Coagulase negative Staphylococcus.

**Table 3 nutrients-11-00581-t003:** CT findings of the 10 positive CT scans (10 episodes in five patients).

Pts	Isolated Pathogens	Non-Nodular Infiltrates	Nodular Opacities	Bilateral Abnormalities	Subpleural Opacities	Cavitations	Pleural Involvment	Pleural Effusion
A	MRSE, vancomycin MIC > 1		x					
B	MRSE, vancomycin MIC > 1	x					x	
B	S chromogenes		x	x	x			
C	MRSA	x				x	x	
C	MRSA	x		x		x		
D	MRSA	x		x				
D	S. haemolytics	x		x				
E	S hominis	x		x				
E	Kocuria kristinae		x	x	x	x		
E	EBSL-producing Pseudomonas aeruginosa	x		x				

CT = Computed Tomography; Pts: patients; A, B, C, D, E are different patients; MRSE = Methicillin-resistant Staphylococcus epidermidis; MIC = Minimal Inhibiting Concentration; ESBL = Extended spectrum beta-lactamase.

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
