# Peer review of "Home Parenteral Nutrition in Patients with Intestinal Failure: Possible Undetected Complications"

_nutrients, 2019, doi:10.3390/nu11030581_

Round 1
Reviewer 1 Report
Please describe clearly the flushing protocol used for the lines while they were not in use -- was it Heparin 300 units? Normal Saline? Taurolidine? Were filters used during infusion of the HPN and was it a 1.2 micron or 0.22 micron filter which would be very stringent at catching actual microbes? This background info needs to be more fully described.
Of the patients who had a line salvaged previously before being enrolled in this corhort, where possible, describe how the line was salvaged, and describe how the patient was then deemed safe to retain that salvaged line in place e.g. repeated blood cultures after 5-7 days of haivng finished systemic antibiotics in combination with antibiotic-lock technique
Please clarify precisely that the echogram performed was an endoscopic esophageal echocardiogram.
Can you describe the morphologies of the peri-catheter thromibi was either valve , flap/tail or other and give the breakdown of number of patients with thrombi conforming to the descriptor. Could this be inlcuded in Table 3?
The premise that Daptomycin because it is ill-suited at treating lung infections, cannot treat septic thromboemboli in the blood vessels of the lung is weak. You need to frame this as a question that it is unknown whether Daptomycin retains it's same efficacy at treating septic thromboemboli in lung vessels as in any other extrapulmonary vessel and this remains an unknown.
Author Response
Dear Reviewer,
attached please find responses to your comments.
Reply to Reviewer 1:
Authors acknowledge the constructive suggestions and comments made by the reviewers. A point by point reply to their questions and criticisms follows below.
Please describe clearly the flushing protocol used for the lines while they were not in use -- was it Heparin 300 units? Normal Saline? Taurolidine?
According to the ESPEN guidelines recommendation (1) central venous access devices were routinely flushed and locked with saline solution when not in use. A new reference (16) has been added.
16. Pittiruti M, Hamilton H, Biffi R, MacFie J, Pertkiewicz M; ESPEN. ESPEN Guidelines on Parenteral Nutrition: central venous catheters (access, care, diagnosis and therapy of complications). Clin Nutr. 2009 Aug;28(4):365-77.
Were filters used during infusion of the HPN and was it a 1.2 micron or 0.22 micron filter which would be very stringent at catching actual microbes? This background info needs to be more fully described.
During HPN infusion, a hydrophobic antibacterial filter with a porosity of 0.4 micron was routinely used. The antibacterial filter was placed on the air intake valve.
This information has been added into the text (page …. line …).
Of the patients who had a line salvaged previously before being enrolled in this cohort, where possible, describe how the line was salvaged, and describe how the patient was then deemed safe to retain that salvaged line in place e.g. repeated blood cultures after 5-7 days of having finished systemic antibiotics in combination with antibiotic-lock technique
According to a consolidated protocol, following the culture results infected catheters were treated with systemic intravenous antibiotic and “lock therapy”; in details, the antibiotic with the lower minimal inhibiting concentration (MIC), after dilution with 5 – 7 mL of saline solution to completely fill the catheter lumen, was injected into the catheter once or two times a day, according to the antibiotic half-life, for a total duration of two weeks. For systemic intravenous therapy a second antibiotic was chosen. The infusion of the nutritional mixture through the catheter was restored at the fourth day of antibiotic therapy. Recurrence of fever during antibiotic lock administration, at the catheter use restoration or at the end of the antibiotic treatment, was considered as therapy failure and indication to the CVC removal.
Please clarify precisely that the echogram performed was an endoscopic esophageal echocardiogram.
At the time of the diagnosis of CVC infection, transthoracic echocardiography, venous vascular echo-colour-Doppler and chest high-resolution CT scans were performed in all patients. Endoscopic esophageal echocardiogram was performed only in cases of uncertain transthoracic ones.
Can you describe the morphologies of the peri-catheter thrombi was either valve, flap/tail or other and give the breakdown of number of patients with thrombi conforming to the descriptor. Could this be inlcuded in Table 3?
In 4 cases the thrombus was on the vessel wall, surrounding the catheter external surface and partially occluding the vein; in 2 cases the thrombus completely occluded the catheter tip.
The premise that Daptomycin because it is ill-suited at treating lung infections, cannot treat septic thromboemboli in the blood vessels of the lung is weak. You need to frame this as a question that it is unknown whether Daptomycin retains it's same efficacy at treating septic thromboemboli in lung vessels as in any other extrapulmonary vessel and this remains an unknown.
In the conclusion section, the sentence on Daptomycin has been modified as suggested:
“ ………… Unfortunately, its efficacy at treating septic thromboemboli in lung vessels as in any other extrapulmonary vessel is still unknown.
In the presence of SPE caused by MR gram-positive bacteria, the use of other antibiotics could be preferable, according to antibiotic sensitivity tests, patients’ clinical conditions and possible drug side effects”.
English revision has been performed, as requested

Reviewer 2 Report
This is a very interesting study. You had a large percentage of patients who experienced SPE. You stated the limitations quite clearly. I would however suggest that you soften the conclusion. Based on the small sample it is hard to recommend CT scans on all patients that have CBSIs. I believe these results need to be confirmed in larger studies and ad different institutions. However you results to provided evidence that there should be a high suspicion of SPE.
Author Response
Dear reviewer,
attached please find the responses to your comments.
Reply to Reviewer 2:
Authors acknowledge the constructive suggestions and comments made by the reviewers. A point by point reply to their questions and criticisms follows below.
This is a very interesting study. You had a large percentage of patients who experienced SPE. You stated the limitations quite clearly. I would however suggest that you soften the conclusion. Based on the small sample it is hard to recommend CT scans on all patients that have CBSIs. I believe these results need to be confirmed in larger studies and ad different institutions. However you results to provided evidence that there should be a high suspicion of SPE.
The conclusion has now been softened, as properly suggested:
“In conclusion, in the present study a chest CT scan showed septic involvement of the lungs in 43,5% of patients with CRBSIs, in the absence of guiding signs or symptoms.
We speculated that in patients with CRBSIs and a high risk for developing complicated infections, a chest CT scan may contribute to detect possible septic pulmonary involvement, thus preventing treatment failure and other potential complications. These preliminary observations need to be confirmed in larger multicentre studies”.
English revision has been performed, as requested
